# Why did informal sector workers stop paying for health insurance in Indonesia? Exploring enrollees' ability and willingness to pay

Muttaqien Muttaqien[1], Hermawati Setiyaningsih[1], Vini Aristianti[1], Harry Laurence Selby Coleman [2], Muhammad Syamsu Hidayat [1,3], Erzan Dhanalvin[4], Dedy Revelino Siregar[4], Ali Ghufron Mukti[1,4], Maarten Olivier Kok [5,6]*

1 Centre for Health Financing Policy and Health Insurance Management (KPMAK), Universitas Gadjah Mada, Yogyakarta, Indonesia, 2 KIT Royal Tropical Institute, Amsterdam, The Netherlands, 3 Department of Public Health, Universitas Ahmad Dahlan, Yogyakarta, Indonesia, 4 BPJS-Kesehatan, Jakarta, Indonesia, 5 Erasmus School of Health Policy and Management, Erasmus University Rotterdam, Rotterdam, The Netherlands, 6 Department of Health Sciences, Vrije Universiteit Amsterdam, Amsterdam, The Netherlands

* m.o.kok@vu.nl

**Data Availability Statement:** https://dataverse. harvard.edu/dataset.xhtml?persistentId=doi:10.

## Abstract

Indonesia faces a growing informal sector in the wake of implementing a national social health insurance system—Jaminan Kesehatan Nasional (JKN)—that supersedes the vertical programmes historically tied to informal employment. Sustainably financing coverage for informal workers requires incentivising enrolment for those never insured and recovering enrolment among those who once paid but no longer do so. This study aims to assess the ability- and willingness-to-pay of informal sector workers who have stopped paying the JKN premium for at least six months, across districts of different fiscal capacity, and explore which factors shaped their willingness and ability to pay using qualitative interviews. Surveys were conducted for 1,709 respondents in 2016, and found that informal workers' average ability and willingness to pay fell below the national health insurance scheme's premium amount, even as many currently spend more than this on healthcare costs. There were large groups for whom the costs of the premium were prohibitive (38%) or, alternatively, they were both technically willing and able to pay (25%). As all individuals in the sample had once paid for insurance, their main reasons for lapsing were based on the uncertain income of informal workers and their changing needs. The study recommends a combination of strategies of targeting of subsidies, progressive premium setting, facilitating payment collection, incentivising insurance package upgrades and socialising the benefits of health insurance in informal worker communities.

## Introduction

A challenge for Asian countries recently implementing social health insurance is covering and incentivising enrolment for the informal sector; the so-called 'missing middle' [1]. To achieve universal health coverage, countries must design health insurance programmes, historically

7910/DVN/5FW8IW https://doi.org/10.7910/DVN/5FW8IW.

**Funding:** The main funder of the study is BPJS Kesahatan, which is the Social Insurance Administration Organization, which administers the Indonesian national health insurance for the Indonesian government. The funder was involved in the design of the study and supported the data collection process. The funder had no role in data analysis and the decision to publish. The funder had the opportunity to review the manuscript before publication, and did not ask for, or make any changes. The analysis and writing were supported, through academic mentorship, by the Dutch organisation for internationalisation in education (NUFFIC), which forms part of the project to strengthen capacity at The Centre of Health Insurance and Financing (KPMAK), Universitas Gadjah Mada, in addressing universal health coverage in Indonesia. NUFFIC has funded this project on behalf of the Netherlands Ministry of Foreign Affairs. Maarten Kok received additional funding through the Research Excellence and Innovation (REI) grant at Erasmus University Rotterdam, which supports research into Universal Health Coverage.

**Competing interests:** Two authors, ED and DRS, are employees of the Social Security Agency, BPJS, which partially funded the study and provided the secondary data on which the analysis is based. These authors were involved in the design of the study and consulted during the drafting of the discussion. They were not involved in the execution of this study, nor the analysis of data, the interpretation of results or the drafting of the result section and discussion. This does not alter our adherence to PLOS ONE policies on sharing data and materials.

tied to formal employment, that reach out to informal sector employees where the majority of jobs' growth has been [2].

Indonesia confronts these problems in the wake of implementing its own social health insurance system—*Jaminan Kesehatan Nasional* (JKN)—that supersedes the vertical programmes for formal sector employees and, separately, the poor, near-poor and informal sector [3]. JKN ought to insure every Indonesian's health. Sustainably financing this coverage however, faces two challenges: incentivising enrolment for those never insured, and perhaps more worryingly, recovering enrolment among those who once paid but no longer do so [4].

While, in 2017, 69 million (57%) of Indonesia's employed workforce were informal sector [5], only 30 million of these are or have been enrolled in JKN previously [6]. Of these 30 million enrolled members, whose premiums are not subsidised by the government, the majority (63.7%) pay the cheapest, entry-level premium (IDR25,500; USD1.96), which entitles them to 'Class III' services, i.e., the basic benefits package [6]. Aside from the 39 million informal sector workers never enrolled, JKN also suffers from a high proportion of non-active informal sector members, meaning those who's premium payment has lapsed. Fourteen million (47%) of the 30 million informal sector workers enrolled are currently non-active, which makes up 75% of the total non-active members in JKN, and represents a substantial revenue base to lose. Finally, there are also informal sector workers who make up an unknown portion of the 120 million *Penerima Bantuan Iuran* (PBI); members whose premiums are subsidised by the government [6]. Given that a significant number of those who have never enrolled, or once did, are likely to qualify for PBI [7], there is a need for Indonesia's health ministry to distinguish between those able to pay but not willing, and those willing to pay but not able.

'Ability to pay' (ATP) and 'willingness to pay' (WTP) are economic concepts that have wide use in the valuation of healthcare [8, 9]. In recent years, both ATP and WTP have been studied in a variety of low- and middle-income countries who are introducing a form a social health insurance, including Ghana, Nigeria, Laos, Pakistan, Ethiopia and Uganda [10–16]. A comparative study hasn't been performed in Indonesia, where JKN constitutes the largest social health insurance programme in the world. These methods are intended to elicit a monetary amount with which an individual would be willing to trade for a specified service or product. In the current case, ATP and WTP can be used to determine whether informal sector workers stop paying premiums due to their personal financial capacity, or whether they lapse due to other factors related to its payment (e.g. low perceived benefit, payment obstacles).

The informal sector contributed just 9% (IDR4.68 trillion or USD359.8 million) to total JKN revenue in 2015, yet accounted for 29% of the total JKN claims, or IDR16.7 trillion (USD1.28 billion) (BPJS 2016, pers. comm., 15 August). There are an estimated 39 million informal sector workers who have never enrolled in JKN, which, crudely assuming each pays a Class III premium, amounts to IDR994.5 billion (USD 76.45 million) revenue stream missed. With total healthcare expenditure far outstripping revenue each year, the gains in coverage and equity in access to healthcare services may be overturned [1]. Aside from financial sustainability, there are arguments for bridging the enrolment gaps based on differences in the access to healthcare services between formal and informal workers [17], the financial protection it offers from excess health expenditure [18], and the governments' own national targets for complete JKN enrolment in 2019.

This study aims to assess the ATP and WTP of informal sector workers who have stopped paying the JKN premium for at least six months, and explore which factors shaped their willingness and ability to pay. Using mixed methods, this assessment can form the basis of policy recommendations which considers the current level of JKN premium, the income level for premium subsidisation and removing the non-financial barriers to continued JKN enrolment.

## Methods

The study was approved by the institutional review board of the faculty of medicine of Universitas Gadjah Mada. Approval number: KE/FK/0232/EC/2019.

### Study design and sample

This is a mixed methods study which uses a sequential explanatory design. A cross-sectional survey, administered to informal sector JKN enrollees (current and former) across 36 cities/districts and 12 provinces, was followed by qualitative interviews among 53 respondents of the same sample.

The sampling method for the quantitative arm of the study was based on a list of registered JKN members from 12 provinces working in the informal sector, provided by BPJS. This list only included Class III premium members who had failed to pay their premium for at least six months. In Indonesia, employees are defined as working in the informal sector if, either, they are self-employed, temporary workers, family workers, or; receiving non-financial family assistance, such as food and board [5]. Based on this list of 926,223 people, a sample size calculation of 1,800 respondents was determined given anticipated effect size. Within this, districts would be purposively selected to achieve a stratified sample of 800 respondents from high fiscal capacity districts, 550 from medium and 450 from low. Provinces and districts in Indonesia are categorised by their fiscal capacity, stipulated in Regulation of the Minister of Finance of the Republic of Indonesia No. 37 of 2016 namely, very high, high, medium and low level. District fiscal capacity is calculated as the total district budget divided by number of poor people (see the online supplementary files for a full description of the calculation of fiscal capacity and the fiscal capacity index). For the purposes of this research, high and very high fiscal capacity districts have been combined due to the small number of very high fiscal capacity provinces. In total, 12 high fiscal capacity districts were sampled, 12 medium and 10 low (Table 1). In each district, a random sample of 55 participants was selected, using Stata's pseudo-random number generator. Thus, 165 respondents were sampled per province (Fig 1).

Two random samples were generated for the qualitative interviews. Each of these was a clustered, random sample in line with the distribution of informal sector workers across high, medium and low fiscal capacity districts. First, based on feasibility considerations, a sample of 25 respondents was generated using a pseudo-random number function. Following this, a second sample of 50 respondents was generated in the same fashion.

### Data collection

Quantitative data were collected using a custom survey questionnaire, developed from several sources. The questionnaire comprised questions on respondents' demographic and socioeconomic characteristics, expenditure, ATP and WTP, the sources of financing for payments, inpatient and outpatient utilisation, healthcare service satisfaction, and reasons for JKN membership. Income and expenditure were measured using questions of Indonesia's national socioeconomic survey [19]. The questionnaire was trialed in one district before being administered across the remaining 33 districts.

ATP was measured in this study using expenditure as a proxy for income, which is relevant for the survey of the informal sector given the difficulty estimating their own income. ATP is usally determined in relation to absolute levels of wealth or assets, and expressed as a percentage of income or expenditure: 2–5% in developing countries [20]. Specifically, ATP was calculated as five percent of a household's non-subsistence expenditure, i.e., less the amount spent on food. This substitution attempts to account for the fewer resources that poorer households can devote to non-food needs [21]. The five percent cut-off derived from measures of typical

**Table 1. Number of respondents per district and province, with each districts' fiscal capacity index.**

| Provincial fiscal capacity | Province | District/city | District Fiscal Capacity | District fiscal capacity index[a] | Respondents (n) |
|---|---|---|---|---|---|
| High | Riau | Siak (d) | High | 3.47 | 51 |
| | | Dumai (c) | High | 1.93 | 38 |
| | | Kuantan Singingi (d) | Medium | 0.93 | 54 |
| | East Kalimantan | Berau (d) | High | 7.57 | 15 |
| | | Penajam Paser Utara (d) | High | 3.15 | 55 |
| | | Samarinda (c) | High | 1.72 | 55 |
| | West Papua | Manokwari (c) | Low | 0.30 | 38 |
| | Papua | Sarmi (d) | High | 5.04 | 45 |
| | | Jayapura (c) | Medium | 0.77 | 60 |
| | | Biak Numfor (d) | Low | 0.44 | 28 |
| | Bali | Badung (d) | High | 6.41 | 54 |
| | | Gianyar (d) | High | 1.20 | 53 |
| | | Buleleng (d) | Medium | 0.63 | 53 |
| Medium | Banten | Tangerang Selatan (c) | High | 9.19 | 55 |
| | | Serang (d) | Medium | 0.59 | 50 |
| | | Pandeglang (d) | Low | 0.22 | 49 |
| | North Sulawesi | Manado (c) | High | 1.17 | 56 |
| | | Minahasa Utara (d) | Medium | 0.74 | 54 |
| | | Minahasa (d) | Low | 0.48 | 54 |
| Low | East Java | Surabaya (c) | High | 1.09 | 54 |
| | | Sidoarjo (d) | Medium | 0.64 | 55 |
| | | Sumenep (d) | Law | 0.13 | 52 |
| | East Nusa Tenggara (NTT) | Ngada (d) | Medium | 0.57 | 50 |
| | | Kupang (c) | Low | 0.41 | 52 |
| | | Rote Ndao (d) | Low | 0.21 | 57 |
| | Central Sulawesi | Palu (c) | Medium | 0.82 | 54 |
| | | Donggala (d) | Low | 0.32 | 49 |
| | | Tojo Una-Una (d) | Low | 0.23 | 53 |
| | North Sumatra | Deli Serdang (d) | Medium | 0.51 | 55 |
| | | Medan (c) | Medium | 0.51 | 55 |
| | | Langkat (d) | Low | 0.24 | 53 |
| | West Kalimantan | Singkawang (c) | High | 1.08 | 51 |
| | | Kubu Raya (d) | Medium | 0.70 | 50 |
| | | Sambas (d) | Low | 0.39 | 52 |

[a]District fiscal capacity: $x \leq 0.5$ = low; $0.5 < x < 1.0$ = medium; $x \geq 1$ = high.

healthcare spending, as a proportion of household income, in developing countries. While this cut-off appears an arbitrary estimate of affordability, it is still half of the most stringent measure of catastrophic health expenditure i.e., when 10% of household expenditure is made up by healthcare spending [22].

WTP was assessed using a bidding game method, where respondents were first asked if their current premium payment was affordable. If respondents' current premium was deemed unaffordable, the amount would be incrementally decreased by 10% until its value was deemed affordable [8, 9, 23]. How data were captured and subsequently coded in the analysis is available in the supplementary online files.

A team of 36 interviewers, each from a single district, was trained in administering the quantitative survey during a day's workshop at Universitas Gadjah Mada, Yogyakarta. The list

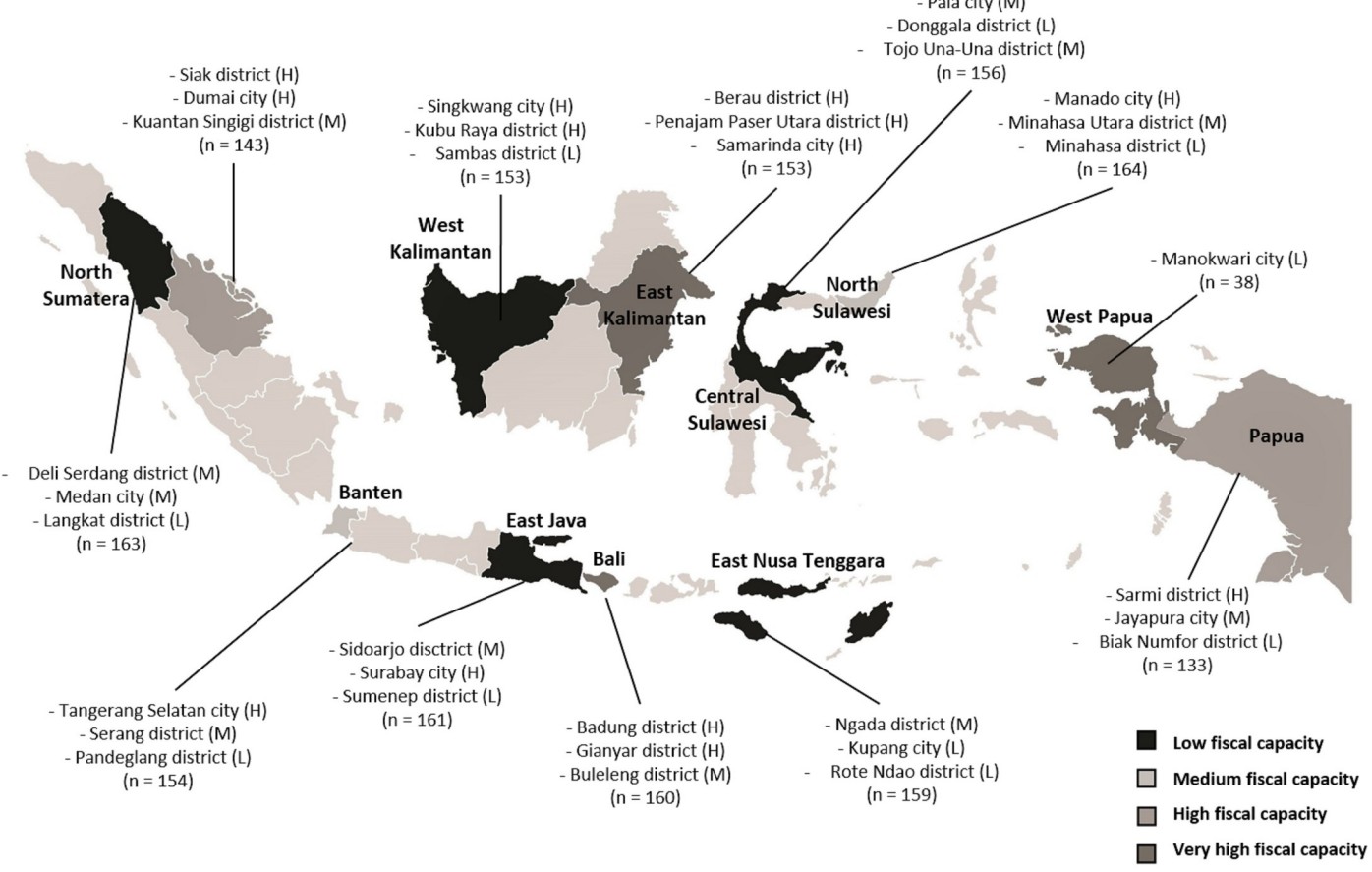

**Fig 1. Districts in which data were collected.**

of informal sector JKN members included complete address which allowed interviewers to perform door-to-door administration of the survey. Participants had to: be 18 years of age or above; be working in the informal sector with a non-salaried occupation, be previously registered for the Class III JKN premium; have not paid the premium for at least six months, and; be non-PBI. The survey was administered between September to November 2016.

For the qualitative study, semi-structured interview guides were developed which sought to understand: respondents' insurance history, their reasons for JKN registration, their length of registration and use of covered services (including any OOP payments), perceived (dis-)advantages of coverage, their payment methods, the reasons for lapsed payment, and potential incentives for re-registration. The interviews were conducted by telephone, by two interviewers: the first author and a research assistant, both of whom are Indonesian males with a background in public health Before each interview, respondents were informed about the aims of the study and content they can expect to discuss. Verbal consent was sought to record the audio from the telephone interviews, which were transcribed verbatim and anonymised to prevent identification.

## Data entry and analysis

The data collected during the survey was entered by three operators and member checks were performed for data validation. The data analysis employed the statistical software package

Stata 12.0 (StataCorp, College Station, TX, USA). Due to the presence of extreme values in the expenditure data, results are reported using both means and medians. T-tests were used to assess the difference between the means or medians for household expenditure between grouped high fiscal capacity districts (HFCD), medium fiscal capacity districts (MFCD) and low fiscal capacity districts (LFCD). As described, ATP was calculated in this study as 5% of total non-subsistence household income, which is total household expenditure minus expenditure for both essential and non-essential food needs. Per capita expenditure and ATP were calculated by dividing by the number of household residents. The influence of ATP and other healthcare utilisation and perception characteristics on WTP was modelled using multivariate regression analyses. These regressions were built based on hypothesised associations between WTP and surveyed personal characteristics and refined by iterative removal of non-significant explanatory variables.

## Results

We found that informal workers' healthcare expenditure on average exceeds the premium amount (IDR 25,500), even as their ATP falls below this level. Expenditures increased with rising district fiscal capacity, which signified greater ATP, but the same trend was not observed between willingness and district wealth. We categorised individuals, based on the frequency distributions of ATP and WTP, into groups that distinguished ATP or WTP as their main constraint to health insurance demand. A large group was 'unable to pay' (38%), while conversely, another was technically willing and able (25%). Questionnaire responses on the reasons for lapsed payment and qualitative interviews suggested informal workers' irregular incomes and changing needs were driving their inconsistent health demand.

### Description of the sample

Surveys were completed for 1,709 respondents, whose demographic and socioeconomic characteristics are shown in Table 2. The occupation of respondents includes: street vendors, labourers, pensioners, farmers, fishermen, handymen, honourers, private employees, entrepreneurs, housewives, clerics, the unemployed and drivers (An honourer is an individual paid an honorarium: a payment made to a volunteer, with no legal obligation on the part of the payer for services received). Among those respondents, 582 participants originate from high fiscal capacity districts, 644 from medium and 483 from low.

Respondents across the three district income groups are relatively homogenous. The sex ratio shifts in low fiscal capacity districts towards women, while medium and low fiscal capacity districts possess slightly higher proportions of individuals with higher education degrees. For occupation, farmers comprise the largest group in low fiscal capacity districts compared to entrepreneurs in medium and high fiscal capacity districts. Although houses of less than four persons comprise a larger proportion in HFCD compared to MFCD and LFCD, average house size is 4.16, 4.02 and 3.95 respectively.

### Healthcare as a proportion of expenditure

Respondents were asked to estimate their monthly expenditure on: essential and non-essential food, clothes, healthcare, and health-related costs (e.g., cigarettes, alcohol). As households are slightly larger on average in HFCD, they will have less to spend on average per capita relative to MFCD and LFCD, where family size is smaller.

Table 3 shows that total household and per capita expenditure increase in higher fiscal capacity districts. There is an inverse relationship, however, between fiscal capacity and healthcare expenditure, as low fiscal capacity districts report the highest healthcare spending: this

**Table 2. Demographic characteristics of the respondents across district fiscal capacity.**

| Variable | High | | Medium | | Low | | Total | |
|---|---|---|---|---|---|---|---|---|
| | n = 582 | | n = 644 | | n = 483 | | n = 1,709 | |
| | n | % | n | % | n | % | n | % |
| **Gender** | | | | | | | | |
| Male | 242 | 41.58 | 292 | 45.34 | 265 | 54.87 | 799 | 46.75 |
| Female | 340 | 58.42 | 352 | 54.66 | 218 | 45.13 | 910 | 53.25 |
| **Education** | | | | | | | | |
| Incomplete primary | 50 | 8.59 | 46 | 7.14 | 46 | 9.52 | 142 | 8.31 |
| Graduate primary | 97 | 16.67 | 104 | 16.15 | 121 | 25.05 | 322 | 18.84 |
| Graduate junior high | 116 | 19.93 | 113 | 17.55 | 76 | 15.73 | 305 | 17.85 |
| Graduate secondary | 263 | 45.19 | 323 | 50.16 | 190 | 39.34 | 776 | 45.41 |
| Diploma | 25 | 4.30 | 15 | 2.33 | 15 | 3.11 | 55 | 3.22 |
| Bachelor's | 29 | 4.98 | 40 | 6.21 | 32 | 6.63 | 101 | 5.91 |
| Master's | 1 | 0.17 | 3 | 0.47 | 1 | 0.21 | 5 | 0.29 |
| PhD | 1 | 0.17 | 0 | 0 | 2 | 0.41 | 3 | 0.18 |
| **Occupation[a]** | | | | | | | | |
| Street Vendor | 11 | 1.95 | 9 | 1.50 | 10 | 2.18 | 30 | 1.85 |
| Labourer | 50 | 8.85 | 51 | 8.51 | 39 | 8.52 | 140 | 8.63 |
| Pensioner | 13 | 2.30 | 19 | 3.17 | 5 | 1.09 | 37 | 2.28 |
| Farmer | 56 | 9.91 | 105 | 17.53 | 141 | 30.79 | 302 | 18.62 |
| Fisherman | 16 | 2.83 | 10 | 1.67 | 11 | 2.40 | 37 | 2.28 |
| Handyman | 39 | 6.90 | 46 | 7.68 | 21 | 4.59 | 106 | 6.54 |
| Honourer | 28 | 4.96 | 16 | 2.67 | 22 | 4.80 | 66 | 4.07 |
| Private employee | 116 | 20.53 | 93 | 15.53 | 46 | 10.04 | 255 | 15.72 |
| Entrepreneur | 162 | 28.67 | 184 | 30.72 | 93 | 20.31 | 439 | 27.07 |
| Housewife | 20 | 3.54 | 7 | 1.17 | 13 | 2.84 | 40 | 2.47 |
| Student | 2 | 0.35 | 0 | 0.0 | 1 | 0.22 | 3 | 0.18 |
| Cleric | 0 | 0.0 | 0 | 0.0 | 6 | 1.31 | 6 | 0.37 |
| Unemployed | 11 | 1.95 | 24 | 4.01 | 12 | 2.62 | 47 | 2.90 |
| Driver | 41 | 7.26 | 35 | 5.84 | 38 | 8.30 | 114 | 7.03 |
| **Household size** | | | | | | | | |
| ≤ 4 | 431 | 74.05 | 460 | 71.43 | 344 | 71.22 | 1,235 | 72.26 |
| > 4 | 151 | 25.95 | 184 | 28.57 | 139 | 28.78 | 474 | 27.74 |

Low is low fiscal capacity, Medium is medium fiscal capacity, High is high fiscal capacity.

ranges from 5.4% of total household expenditure in low fiscal capacity districts, to 3.6% and 2.8% in medium and high, respectively. As such, low fiscal capacity districts report spending more on healthcare than the 2016 price of the JKN premium at the time. The proportion of total household income consumed by health spending across districts was 6.2% in LFCD, 3.7% in MFCD and 2.7% in HFCD: a range which closely resembles Russel's previous findings for household ATP in LMICs.

## Ability to pay

Ability to pay is calculated here as 5% of total non-subsistence effective household expenditure, and reported per capita. By dividing by household size, this calculates a proportion of total household income that is available for each person in that house under the allocation of health. Given informal sector workers may face difficulties accurately estimating their own income,

**Table 3. The mean income and expenditure per household and per capita (IDR 000s) across districts.**

| District fiscal capacity | Mean household size | Mean Income (SD) | | Mean expenditure (SD) | | Mean healthcare expenditure (SD) | | | Mean health insurance expenditure (SD) | | |
|---|---|---|---|---|---|---|---|---|---|---|---|
| | | ph | pc | ph | pc | ph | pc | % of total ph (pc) | ph | pc | % of total ph (pc) |
| Low | 3.95 | 1,708 (1,283) | 500 (449) | 1,928 (1,106) | 555 (370) | 105 (356) | 33 (112) | 5.43% (5.96%) | 23 (106) | 6 (25) | 1.19% (1.12%) |
| Med | 4.02 | 2,470 (1,729) | 660 (485) | 2,523 (1,410) | 675 (408) | 91 (488) | 23 (101) | 3.62% (3.40%) | 42 (161) | 11 (40) | 1.67% (1.58%) |
| High | 4.16 | 2,820 (1,885) | 720 (514) | 2,697 (1,488) | 678 (373) | 75 (300) | 19 (79) | 2.80% (2.86%) | 51 (21) | 11 (40) | 1.87% (1.63%) |
| Total | 4.05 | 2,374 (1,730) | 635 (493) | 2,414 (1,395) | 643 (389) | 91 (401) | 25 (99) | 3.78% (3.94%) | 37 (159) | 9 (36) | 1.56% (1.44%) |
| Min | 1 | 0 | 8 | 75 | 25 | 0 | 0 | - | 0 | 0 | - |
| Max | 13 | 14,000 | 4,500 | 9,042 | 3,754 | 10,000 | 2,000 | - | 3,000 | 750 | - |

General healthcare and health insurance expenditures are also shown as a proportion of total expenditure.

ph = per household; pc = per capita.

respondents were asked to estimate both income as well as expenditure. Indeed, this is evidenced by the higher and lower bounds of income shown in Table 3 above, and the consistently lower estimations of income than expenditure.

Fig 2 shows the frequency distribution of ATP across the sampled informal sector workers. Over 80% (n = 1,378) of individuals have an ATP below the premium amount—IDR25,500—while 72% (n = 349) of individuals from LFCD have an ATP below the average ATP for the sample: IDR16,571 (USD1.27). Table 4 shows that, across districts, ATP rises alongside fiscal capacity; from IDR13,480 (USD1.04) in LFCD, IDR17,497 (USD1.35) in MFCD and

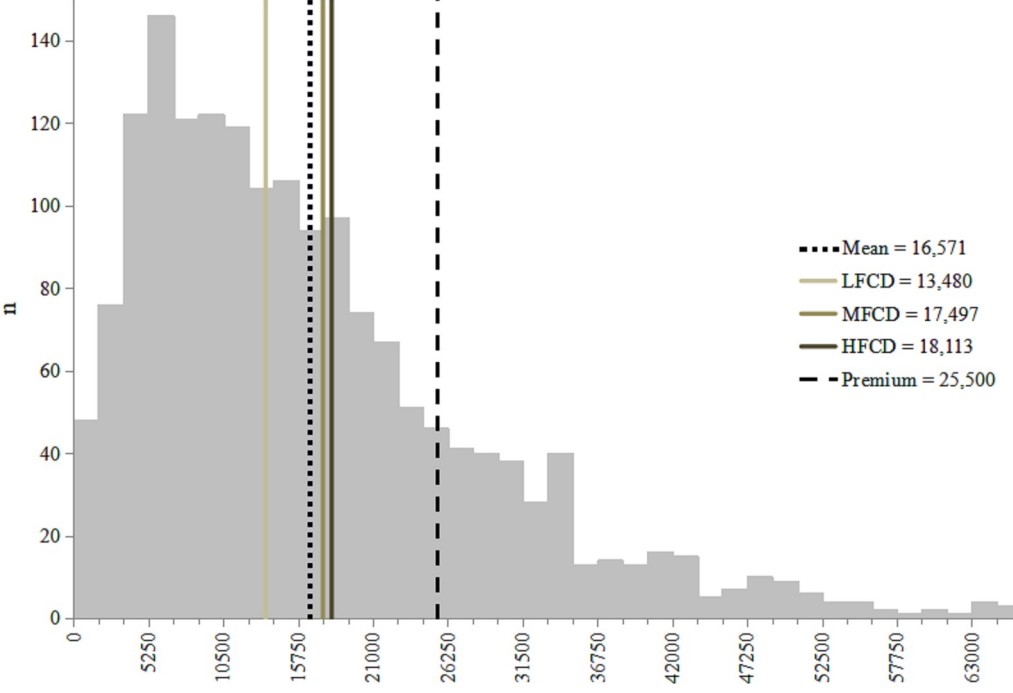

**Fig 2. Ability to pay.**

**Table 4. Ability to pay per capita across district fiscal capacity.**

| District fiscal capacity | Mean ATP (SD) | Median | Min | Max |
|---|---|---|---|---|
| Low (n = 483) | 13,480 (10,892) | 10,625 | 347 | 61,000 |
| Med (n = 644) | 17,497 (12,083) | 14,508 | 350 | 65,767 |
| High (n = 582) | 18,113 (11,895) | 16,575 | 250 | 64,120 |
| Total (n = 1,709) | 16,571 (11,850) | 13,991 | 250 | 65,767 |

Low: low fiscal capacity; Med: medium fiscal capacity; High: high fiscal capacity.

IDR18,113 (USD1.39) in HFCD. Thus, ATP per capita in all three district income groups is lower than Class III premium. This premium amount would consume 7.0% in HFCD, 7.3% in MFCD and 9.5% in LFCD of total non-subsistence effective expenditure, and, respectively, 3.4%, 3.7% and 4.9% of per capita income.

## Willingness to pay

We used a bidding game to determine respondents' maximum WTP. The game started by asking respondents' if the costs of JKN's class III premium was affordable. This amount was either decreased or increased by 10% increments depending on a negative or positive response, until the cut-off of their 'willingness' was found i.e., the first lowest/highest amount at which they positively respond.

The distribution of WTP can be seen in Fig 3. Twenty-four percent of respondents lie above the level of the current premium, thus indicating their WTP should satisfy the needs of JKN enrolment. Thirty-four percent of those below the mean WTP are from LFCD while the remaining are from HFCD and MFCD. For the entire sample, average WTP was IDR12,485 (USD0.96). This ranges from the lowest WTP in LFCD of IDR9,923 (USD0.76), to the highest

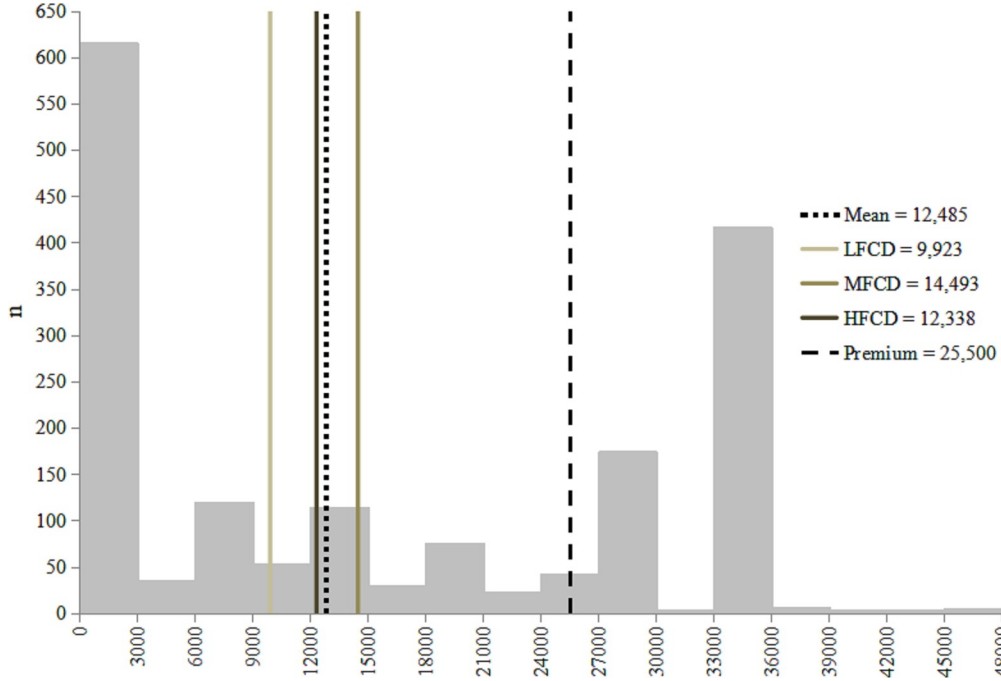

**Fig 3. Willingness to pay.**

**Table 5. Willingness to pay per capita across district fiscal capacity.**

| District fiscal capacity | Mean WTP (SD) | Median | Min | Max |
|---|---|---|---|---|
| Low (n = 483) | 9,923 (11,551) | 5,500 | 0 | 48,000 |
| Med (n = 644) | 14,493 (11,936) | 15,500 | 0 | 48,000 |
| High (n = 582) | 12,388 (11,502) | 10,500 | 0 | 48,000 |
| Total (n = 1,709) | 12,485 (11,818) | 10,500 | 0 | 48,000 |

Low: low fiscal capacity; Med: medium fiscal capacity; High: high fiscal capacity.

in MFCD of IDR14,493 (USD1.11). Thus, WTP was consistently lower than ATP and substantially lower than the JKN premium in 2016 (Table 5).

## ATP and WTP

Figs 2 and 3 show individuals may face ATP as their main constraint (i.e., do not have the financial capacity to make premium payment) or WTP (i.e., they do not perceive the payment as worth making), or both. For both ATP and WTP, individuals were recategorized into two groups on the basis of whether ATP or WTP was their main constraint to payment. The ranges used to determine these groups are shown below.

"Unable" = 0≤x<10,625
"Able" = x≥10,625
"Unwilling" = 0≤x<25,500
"Willing" = x ≥ 25,500

The JKN class III premium amount–IDR25,500–denotes the amount at which respondent's income should no longer be a constraint to their ATP or WTP. However, IDR10,625 (USD0.82) is the median ATP of low fiscal capacity districts i.e., for 50% of individuals in low fiscal capacity districts, 5% of their non-subsistence effective income lies above 10,625. This value was chosen to distinguish individuals clearly unable to pay and those 'nearly able'; i.e., the latter may also benefit from interventions that don't solely relate to their financial capacity.

Table 6 shows the mean ATP for the groups formed above. This shows that 25% of the sample are both willing to pay above the JKN premium, and their financial capacity is sufficient to do so. There is another group, comprising 38% of the sample, whose ATP falls substantially below the level of the premium. This marks the distinction between individuals who mainly lapse in premium payment due to financial capacity, and those who fail to pay because of their perceptions of JKN enrolment.

## Determinants of ATP and WTP

Table 7 shows the final models of the multiple logistic regression, using the groups 'unable to pay' ($R^2 = 0.19$) and 'willing to pay' ($R^2 = 0.07$) as the outcome variables of separate analyses.

**Table 6. Mean ATP for groups with ATP or WTP as their predominant constraining factor.**

| | n (%) | Mean ATP (SD) | Min | Max |
|---|---|---|---|---|
| Unable | 644 (38%) | 6,125 (2,704) | 250 | 10,613 |
| Able | 1,065 (62%) | 22,888 (10,723) | 10,625 | 65,767 |
| Unwilling | 1,280 (75%) | 7,205 (8,579) | 0 | 25,000 |
| Willing | 429 (25%) | 28,239 (2,303) | 25,500 | 48,000 |

Unable = 0≤x<10,625; Able = x≥10,625; Unwilling = 0≤x<25,500; Willing = x ≥ 25,500.

**Table 7. Determinants of being unable to pay, and willing to pay (i.e., ATP lies below 10,625, WTP lies above 25,500).**

| | Unable to pay | Willing to pay |
|---|---|---|
| **Variable** | OR (95% CI) | OR (95% CI) |
| **Age** | 1.01* (1.00–1.02) | 0.98* (0.98–0.99) |
| **Household size** | 1.24** (1.13–1.35) | 1.18** (1.08–1.29) |
| **Urban district[a]** | - | 1.42** (1.10–1.84) |
| **Income per capita** | 0.99** (0.99–1.00) | 1.00** (1.00–1.00) |
| **District fiscal capacity** | | |
| **Medium fiscal capacity** | 0.63** (0.48–0.83) | 1.41* (1.05–1.90) |
| **High fiscal capacity** | 0.68* (0.52–0.91) | 0.67* (0.48–0.93) |
| **Willing[b]** | 0.59** (0.45–0.78) | - |
| **Unable to pay[c]** | - | 0.54** (0.41–0.71) |
| **ref** | 0.94 (0.52–1.70) | 0.20** (0.11–0.35) |

[a]0 = Rural district; 1 = Urban district
[b]'Willing': 0 = (x < 25,500); 1 = (x ≥ 25,500)
[c]'Unable to pay': 0 = x ≥ 10,625; 1 = x < 10,625
*$p \leq 0.05$
**$p \leq 0.01$.

Aside from the perfect linear relationship between income and ATP or WTP, household size is positively associated with a higher odds of being both unable to pay (OR = 1.24; 95%CI: 1.13, 1.35), and willing to pay (OR = 1.18; 95%CI: 1.08, 1.29). Increasing age was associated with a small effect on inability to pay (OR = 1.01; 95%CI: 1.00, 1.02), and an inverse effect willingness to pay (OR = 0.98; 95%CI: 0.98, 0.99), and there was a reciprocal association between these outcome variables of reduced odds when predicting each other. Being 'willing to pay' was associated with a 41% reduction in the likelihood of being unable to pay (95%CI: 0.45, 0.78), and conversely, being unable to pay was associated with 46% reduced odds of being willing (95% CI: 0.41, 0.71). Compared to LFCD, being from a MFCD (OR = 0.62; 95%CI: 0.48, 0.83) or HFCD (OR = 0.68; 95%CI: 0.52, 0.91) reduced the odds of being unable to pay, yet MFCD had increased odds of being willing (OR = 1.41; 95%CI: 1.05, 1.90) and lower odds in HFCD (OR = 0.67; 95%CI: 0.48–0.93).

## Reasons for lapsed payment

The dominant reasoning for lapsed JKN payments, across the four groups constructed above, was also evaluated during the survey (Table 8). This reasoning is relatively consistent across groups, with income uncertainty the most commonly cited reason for lapsed payment for all but willing groups. Difficulties associated with payment and forgetting to pay were also common across groups, while being disappointed with BPJS healthcare, or healthcare provision in general, ranked consistently low in respondents' priorities.

## Qualitative results

An in-depth understanding of why those who were recently willing and able to pay had stopped doing so was provided by the qualitative interviews. Fifty-three respondents, were interviewed over the phone. The sample was predominantly male, with an average age of 42, family size of four and over 64% had received some form of secondary school education. Street vendors, farmers, labourers and private employees accounted for 72% of the sample. Only

**Table 8. Reasons for lapsing in JKN premium payment.**

| Reason | Able (n = 820) | | | Unable (n = 528) | | | Willing (n = 331) | | | Unwilling (n = 1,017) | | | Total (n = 1,348) | | |
|---|---|---|---|---|---|---|---|---|---|---|---|---|---|---|---|
| | n | (%) | Rank | n | (%) | Rank | n | (%) | Rank | n | (%) | Rank | n | % | Rank |
| Income uncertainty | 288 | (35) | 1 | 249 | (47.2) | 1 | 65 | (20) | 2 | 472 | (46.4) | 1 | 537 | (39.8) | 1 |
| Conditions of insured BPJS services | 46 | (6) | 7 | 25 | (4.7) | 7 | 23 | (7) | 7 | 48 | (4.7) | 7 | 71 | (5.3) | 7 |
| Disappointment with healthcare services | 79 | (10) | 5 | 31 | (5.9) | 6 | 42 | (13) | 4 | 68 | (6.7) | 6 | 110 | (8.2) | 5 |
| Disappointment with BPJS-Health | 68 | (8) | 6 | 38 | (7.2) | 5 | 33 | (10) | 6 | 73 | (7.2) | 5 | 106 | (7.9) | 6 |
| Difficulties associated with payment | 107 | (13) | 3 | 68 | (12.9) | 2 | 55 | (17) | 3 | 120 | (11.8) | 3 | 175 | (13.0) | 3 |
| Forget to pay | 133 | (16) | 2 | 51 | (9.7) | 4 | 74 | (22) | 1 | 110 | (10.8) | 4 | 184 | (13.6) | 2 |
| Other | 99 | (12) | 4 | 66 | (12.5) | 3 | 39 | (12) | 5 | 126 | (12.4) | 2 | 165 | (12.2) | 4 |

Unable = 0≤x<10,625; Able = x≥10,625; Unwilling = 0≤x<25,500; Willing = x ≥ 25,500.

three respondents had previously registered for another form of insurance, suggesting respondents' difficulties paying were not reserved for JKN.

The most common reason cited for lapsed payment was income uncertainty, or its causes by seasonal and temporary work. A private employee from a LFCD with an ATP of IDR 8,950 (USD 0.69) explained:

> *We are not civil servants, so I don't have [a fixed salary] [. . .] Here again, I am in arrears and have not paid yet. [I am] waiting till next month if there is money as I want to pay all [the arrears].* **R1**

Individuals across district income groups and financial capacities had experienced difficulties reconciling their irregular income with fixed monthly payments. A private employee from a HFCD and an ATP over the premium amount cited the opportunity costs to paying JKN premiums; those of buying rice, water or paying for electricity. Other respondents described being too poor to pay yet not exempted; citing failures in the national registration system for PBI.

Other reasons that respondents cited were forgetting to pay, difficulties associated with payment and dissatisfaction with BPJS or general healthcare services. Tied to income uncertainty, there were numerous responses which detailed the difficulty of paying back arrears, particularly as these accumulate over months. A 44-year old from a family of three and ATP over the premium amount underlined this reasoning behind being unwilling to pay anything for BPJS:

> *Oh, I don't pay anymore. We were diligent in paying at first. Then I was one month late, and I wanted to pay for the late month with [the current month still to pay]. But suddenly, BPJS told me to pay three months at one time, right? [. . .] After one year I intended to pay again, but BPJS wanted me to pay everything immediately, for three people. It's hard for me, because how many millions is that right away?* **R2**

Several respondents had discontinued their JKN registration due to the monthly recurring payments being incompatible with their revenue stream, and there were additional hurdles described with registration or payment that pushed respondents away. Half of respondents did not possess a bank account; thus, several different payment methods were described and these followed what was easiest for each individual. Of the 39 respondents who described payment, 26 (66.67%) had paid by cash at a shop, bank, post office or BPJS office, while ten respondents said that they paid by card over ATM, and three people's premiums were paid by relatives.

We asked respondents why they had registered for BPJS health insurance in the past, and four different reasons were described. Fifteen respondents said that it would allow them to access health care if needed in the future. Another fifteen said that they were following what their relatives were doing, or they were encouraged at a health facility and following national regulations. Twelve people said that they had registered because they knew that they would soon have substantial health spending, i.e. expecting to give birth soon. Finally, seven people emphasised that it would protect them against unexpected health expenditure. Respondents were generally positive about the aims of JKN, as it could reduce future medical costs and made accessing healthcare easier:

> *Based on my opinion, the service is good. But the drugs are not available and they asked us to buy this outside. The problem is that we must buy them with our own money.* **R3**

As with this quote, the suggestions made by respondents on improving the service pointed to healthcare providers gaming the system. Forty-three respondents had used their insurance to access primary or hospital care. Yet, of the 31 respondents who had used inpatient services, 18 (58%) still had to pay out of pocket for medicines. Even though, medicine should be provided free, respondents described paying between IDR200,000 (USD15.38) and 10million (USD768.76) as hospital staff told them that these specific medicines were not covered by JKN. Other restrictions on JKN members were described, such as healthcare providers having a limited number of JKN patients they could see each day, or a limited number of beds for JKN patients. However, beds became available and waiting times could be skipped if respondents decided to pay out of pocket, as one respondent did.

## Discussion

Across districts of varying fiscal capacity, the mean ATP and WTP of informal sector workers was considerably lower than the premium amount (IDR8,929 and 13,015, respectively). That these same individuals had, at one point, registered and paid for JKN suggests that assessments of ATP and WTP are fluid, and subject to change. In fact, individuals from LFCDs and MFCDs were already spending, on average, above or near the premium amount on healthcare costs. This higher general healthcare spending was inversely related to health insurance spending, more common of HFCDs. There are many possible explanations for these trends. Access could be easier in districts with higher healthcare spending, or a larger proportion of that spending could be made up by out-of-pocket payments, arising from less well-resourced health facilities or more fluid needs. A higher number of HFCD districts may have subsidised local health insurance (Jamkesda), or more simply there may be a greater need and demand for services in districts with higher health spending. Previous health evaluation research has suggested Engel's law holds for health spending, where the proportion of income spent on health would decrease as income increases, even as the absolute amount of health spending continued to rise [24]. While our study compares summary measures of income across districts of different wealth, rather than the income at the individual level, it suggests absolute decreases in health spending as income, or district wealth, increases. Second, we find that the informal sector is not a homogenous group in relation to its demand for health. These differences became apparent when contrasting individuals who were predominantly constrained by their finances (i.e., ATP) versus their perceptions (i.e., WTP). The median ATP of LFCD (IDR10,625) was used as the cut-off to distinguish those nearly able to pay versus those clearly unable to pay, and, as ATP is derived from expenditure, it represents the poorest 50% of individuals from the poorest districts. In practice, this can be used to highlight individuals for whom the costs of the

premium are prohibitive, which was 38% of the sample. Research in China showed that, among those with low contribution levels, health insurance coverage steeply declined as premium costs increased, suggesting an elastic demand for health [25]. As the logistic regression in this study shows WTP increases as ATP increases, and thus, people change their willingness in line with their own finances, subsidising premiums for those unable to pay should be an effective measure for increasing demand. Aside from this association, we also found that age, household size and living in urban areas affected ATP, WTP or both. Similar effects of age and household size on WTP have also been identified in other contexts [25–27], while other studies have identified a lower WTP in districts compared to cities [28], and communities living further from health facilities [29, 30]. A possible explanation for this finding is in the suggested effect of increased supply on raising WTP [25], i.e., the better resourced and more easily accessible health facilities in urban districts [31] create greater demand compared to poorly resourced facilities in rural areas. Finally, 25% of this sample are both technically willing and able to pay (i.e., both amounts lie above the premium), even though the sample only includes people who have missed six months of payments. This points to a dynamic nature of ATP and WTP, which we explored further through evaluating individuals' reasons for lapsed payments and more in-depth qualitative interviews.

The responses of informal sector workers suggested the main problem they face in reconciling their ability and willingness to pay were their fluctuating income and needs. Almost consistently ranked across districts as the main reason for failing to pay their premiums, the uncertainty of informal sector incomes was deemed incompatible with the monthly recurring JKN payment. Research on informal workers in Bangladesh suggested occupations with daily wages (such as, rickshaw pullers) had a higher WTP than those paid weekly or monthly given the former have more ready access to liquidity [28]. Income uncertainty could also form part of a broader category of payment problems faced by the informal sector, which includes the second and third ranked reasons for lapsed payment: forgetting to pay and difficulties associated with payment. The difficulties linked to the recurrence of premium payments which affected informal sector workers' ability to settle arrears when payments were missed. Having to settle the entire debt in one payment was prohibitive to individuals seeking to recover their insurance status. Further, asking respondents why they had signed up for JKN revealed that a proportion expected significant health spending in the near future. This illustrates the fluid nature of informal workers' WTP, which changes in line with their needs, as ATP does with their finances. This echoes previous research that found insurance was not used by the informal sector to recover the costs of frequent, common illnesses but for the protection it offers against rare but large financial losses from acute episodes [25]. Finally, we found that discontinuing payment was not primarily driven by disappointment with health services, which discounts the idea that low WTP was based on perceptions of JKN as an inferior product. This raises further questions when considered against respondents' suggested improvements to JKN, which mainly described providers gaming services, such as demanding out of pocket payments for medicines, and insisting beds or doctors for JKN patients were occupied. While, apparently, these problems do not lead deterrence, their removal may contribute to changing the incentives driving enrolment: from the 'push' of avoiding financial penalties to the 'pull' of anticipated health service benefits.

The study also possesses a number of limitations. A risk of the bidding game method in the evaluation of WTP is that the initial bid value influences the value of final accepted bid, known as starting point bias. We believe this study transcends this risk given the existing premium amount, which informal sectors workers were paying until six months previously, provides a real-world value rather than a hypothetical starting point [32]. In addition, while the survey was conducted in 2016, real GDP per capita growth from 2016 to 2019 has been relatively

small, given consistent levels of annual GDP growth have corresponded to similar levels of inflation. Thus, there is limited reasons to suspect there have been large changes to the economic circumstances of the informal sector which would change the interpretation of the results today.

The following recommendations are grounded in the understanding that informal sector workers may face constraints in meeting the premium primarily due to their ability to pay, their willingness or both. It follows that these different groups will require different interventions in order to achieve universal health coverage of the informal sector. The subsequent recommendations are structured within this framing: whether they aim to raise ability or willingness to pay.

## Recommendations for raising ATP

**PBI targeting, subsidisation and progressive premium setting.** This study suggests that, for a considerable proportion of informal sector workers, the premium would comprise significantly more than 5% of non-subsistence income, which is a typical level of health spending in LMICs. Thus, with incomes too low to reasonably expect meeting this financial demand, the targeting of subsidised premiums (PBI) needs to be re-evaluated. It has been reported that the Finance and Development Supervisory Agency (BPKP) found four million people were double registered for JKN, and the PBI registration system contained false claimants and missed many eligible persons [33]. Regulation defining PBI eligibility describes persons 'unable to pay', yet does not stipulate the income levels at which individuals qualify. This research uses a summary measure of district wealth to define being 'unable to pay', yet we suggest individual measures of income can inform setting multiple levels for PBI that grant the recipient partial or full premium subsidies. Partial subsidies for the informal sector have been successfully employed in China, India, Vietnam, Korea, Japan and Taiwan [34, 35]. As this study indicates willingness changes in line with ATP, the aforementioned measures can be matched with a progressive system of premium setting, tiered in line with ATP, to increase collectability at low-income levels.

## Recommendations for raising WTP

**Facilitating payment collection.** This study suggests that a primary driver of discontinued payment among informal sector workers are the obstacles to making premium and arrears payments. This includes the channels through which payment can be made, their modes and frequency. The former is substantiated by other research linking the time cost and convenience of making premium payments to their likelihood of being paid by informal workers [36, 37]. Further, the government's own research has recommended amending premium payment plans for informal workers that are in line with their irregular cash flow [38]. This study suggests the latter is particularly influential on the likelihood of paying back arrears, where large one-time payments can be expected in order to regain insurance status. While BPJS has already introduced a system of door-to-door collection, called Kader-JKN, research suggests payment collection has been most effective in countries with strong accountability pressures on local governments to achieve payment target [17, 36].

**Incentivising premium upgrading.** In this study, a quarter of informal workers were both willing and able to pay, yet no longer doing so. Aside from enhancing continued enrolment, a proportion of these workers should be incentivised to purchase a more comprehensive insurance package with their heightened ATP. Yet, the JKN system currently allows individuals to 'top-up' coverage for inpatient services at the point of delivery, for example to upgrade on room size and privacy. This may serve as a disincentive for continued payment of a more

comprehensive insurance package over the long term. A good first step has been regulation limiting these 'top-ups' to single levels (i.e., from Class III to Class II), however making an informed assessment of different insurance packages requires a communal conception of insurance, financial risk protection and prepayment for health that comes with the greater promotion and socialisation of JKN.

**Socialisation of JKN.** In 2017, there were 39 million informal workers who had never been registered in JKN (DJSN, 2017). There is empirical evidence showing that the demand for health insurance is contingent on having an understanding of the prospective benefits of that insurance [39]. The above recommendations on PBI eligibility, insurance costs, payment methods and insurance packages require informal workers that can make informed judgements. A review by Dror and Firth discounts that informal workers make individual evaluations of utility and costs, but that decisions are based on group affiliation; thus, demand arises in communities when it "recognises that purchasing health insurance is welfare enhancing and consistent with 'what responsible adults should do'" [40]. They suggest that understanding local governance structures, and designing health communication strategies that make sense within these rules, can drive the establishment of demand in informal worker communities.

Assuming ATP comprises a consistent amount of family or per capita income ignores that households may have different priorities and thus spending patterns to meet their needs. Further, ATP was measured in this study using expenditures, and rests on the assumption that it can indicate demand for health insurance, when as discussed, communities may not recognise it as a valuable product [41]. District fiscal capacity was used in this study as a proxy for income or financial means, which is instructive for policy makers to understand the wider socioeconomic drivers of geographic variations in health insurance demand, but less applicable for recommending income levels for subsidisation. WTP was assessed using a bidding game on which respondents must conceptualise affordability. Although this study found that individuals adjust their conception of affordability in line with rising income and expenditures, the study still assumes that affordability remains within a relatively stable range across groups. Finally, this study looked at informal workers who had missed six months of payments, so these findings may not be generalizable to the 39 million informal workers who have never registered for JKN. Understanding the local governance structures of informal worker communities and how these, and the tacit rules and values they follow, could be consistent with a social conception of insurance would be a valuable subsequent research direction to characterise demand for those who once paid, as well as those who are yet to enroll in JKN.

## Conclusion

This study of informal workers in Indonesia found that, across districts of different fiscal capacity, their average ability and willingness to pay fell below the national health insurance scheme's premium amount. This group could be further sub-categorised based on whether their financial capacity or perceptions of insurance were the main constraint to their demand. There were large groups at both ends for whom the costs of the premium were prohibitive or, alternatively, they were both technically willing and able to pay. As all individuals in the sample had once paid for insurance, their main reasons for lapsing were based on the uncertain income of informal workers and their changing needs. This study recommends a combination of strategies of targeting of subsidies, progressive premium setting, facilitating payment collection, incentivising insurance package upgrades and socialising the benefits of health insurance in informal worker communities.

## Supporting information

**S1 Appendix. Calculation of fiscal capacity and fiscal capacity index.**
(DOCX)

**S1 File. Quantitative questionnaire English.**
(PDF)

**S2 File. Quantitative questionnaire Indonesian.**
(PDF)

**S3 File. Qualitative questions ATP/WTP Indonesian.**
(DOCX)

**S4 File. Qualitative questions ATP/WTP English.**
(DOCX)

## Acknowledgments

We would like to thank the questionnaire and interview respondents for their time and making this research possible. Big thanks to BPJS Kesehatan for providing the secondary data that was invaluable for this research. Also thanks to Esther den Hartog, Diah Ayu Puspandari and the staff of KPMAK, Maria Lastri Sasanti, Dwi Martiningsih, Andi Afdhal Abdullah (BPJS Kesehatan) and Elizabeth Pisani for their support.

## Author Contributions

**Conceptualization:** Muttaqien Muttaqien, Vini Aristianti, Muhammad Syamsu Hidayat, Erzan Dhanalvin, Dedy Revelino Siregar.

**Data curation:** Hermawati Setiyaningsih, Vini Aristianti, Erzan Dhanalvin, Dedy Revelino Siregar.

**Formal analysis:** Muttaqien Muttaqien, Hermawati Setiyaningsih, Vini Aristianti, Harry Laurence Selby Coleman, Muhammad Syamsu Hidayat, Maarten Olivier Kok.

**Funding acquisition:** Muttaqien Muttaqien, Ali Ghufron Mukti.

**Investigation:** Muttaqien Muttaqien, Vini Aristianti, Ali Ghufron Mukti, Maarten Olivier Kok.

**Methodology:** Muttaqien Muttaqien, Vini Aristianti, Harry Laurence Selby Coleman, Muhammad Syamsu Hidayat, Maarten Olivier Kok.

**Project administration:** Maarten Olivier Kok.

**Resources:** Maarten Olivier Kok.

**Software:** Hermawati Setiyaningsih, Vini Aristianti.

**Supervision:** Muttaqien Muttaqien, Muhammad Syamsu Hidayat, Ali Ghufron Mukti, Maarten Olivier Kok.

**Writing – original draft:** Muttaqien Muttaqien, Vini Aristianti, Harry Laurence Selby Coleman, Muhammad Syamsu Hidayat, Maarten Olivier Kok.

**Writing – review & editing:** Muttaqien Muttaqien, Hermawati Setiyaningsih, Vini Aristianti, Harry Laurence Selby Coleman, Muhammad Syamsu Hidayat, Erzan Dhanalvin, Dedy Revelino Siregar, Ali Ghufron Mukti, Maarten Olivier Kok.

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
