## [Decision Letter · Decision Letter 0]

27 Jan 2021

PONE-D-20-30634

Ability and willingness to pay for National Health Insurance in Indonesia: a study of the informal sector

PLOS ONE

Dear Dr. Kok,

Thank you for submitting your manuscript to PLOS ONE. After careful consideration, we feel that it has merit but does not fully meet PLOS ONE’s publication criteria as it currently stands. Therefore, we invite you to submit a revised version of the manuscript that addresses the points raised during the review process.

We look forward to receiving your revised manuscript.

Kind regards,

David Hotchkiss

Academic Editor

PLOS ONE

Furthermore, when reporting the results of qualitative research, we suggest consulting the COREQ guidelines: http://intqhc.oxfordjournals.org/content/19/6/349. In this case, please consider including more information on the number of interviewers, their training and characteristics; and please provide the interview guide used.

"I have read the journal's policy and the authors of this manuscript have the following competing interests: Two authors, ED and DRS, are employees of the Social Security Agency, BPJS, which partially funded the study and provided the secondary data on which the analysis is based. These authors were involved in the design of the study and consulted during the drafting of  the discussion. They were not involved in the execution of this study, nor the analysis of data, the interpretation of results or the drafting of the result section and discussion."

5. We note that Figure 1 in your submission contain map images which may be copyrighted. All PLOS content is published under the Creative Commons Attribution License (CC BY 4.0), which means that the manuscript, images, and Supporting Information files will be freely available online, and any third party is permitted to access, download, copy, distribute, and use these materials in any way, even commercially, with proper attribution. For these reasons, we cannot publish previously copyrighted maps or satellite images created using proprietary data, such as Google software (Google Maps, Street View, and Earth). For more information, see our copyright guidelines: http://journals.plos.org/plosone/s/licenses-and-copyright.

(1) You may seek permission from the original copyright holder of Figure 1 to publish the content specifically under the CC BY 4.0 license. 

Reviewers' comments:

Reviewer's Responses to Questions

**Comments to the Author**

1. Is the manuscript technically sound, and do the data support the conclusions?

Reviewer #1: Partly

Reviewer #2: Partly

2. Has the statistical analysis been performed appropriately and rigorously? 

Reviewer #1: N/A

Reviewer #2: I Don't Know

3. Have the authors made all data underlying the findings in their manuscript fully available?

Reviewer #1: Yes

Reviewer #2: Yes

4. Is the manuscript presented in an intelligible fashion and written in standard English?

Reviewer #1: Yes

Reviewer #2: Yes

5. Review Comments to the Author

Reviewer #1: This paper addressed one of very critical issues of the health insurance coverage among informal sector from a unique angle. The topic and results of this study could be very valuable for JKN development as well as the development of health insurance schemes in other LMICs.

Here are a few recommendations for the potential improvement.

1. Can author add the sections regarding the literature review and theories related to this study? ATP and WTP have been studied intensively. It would be great that authors can show how these existing knowledges guide the design and analysis of this study. For example, we understand that people’s health state will be an important factor affecting their WTP and actual enrollment of insurance significantly.

2. At the end of the paper, authors made some very useful policy recommendations. It would be great if authors can make better connections between the results of study and these recommendations, with the considerations of feasibilities of their implementation in JKN.

3. In line 124, It is stated that “ATP was calculated as five percent of a household’s non- subsistence expenditure, i.e., less the amount spent on food”; while in line 127, it is stated that “it is still half of the most stringent measure of catastrophic health expenditure”. To my knowledge, these two are two different measurements, and please make sure their compatibilities.

Reviewer #2: Referee Report PONE-D-30634

Ability and willingness to pay for National Health Insurance in Indonesia: a study of the informal sector.

The iterative bidding approach (Randall et al., 1974) starts by querying individuals at some initial dollar value and keeps raising (or lowering) the value until the respondent declines (accepts) to pay. This final dollar amount is interpreted as the respondent's WTP. However, this approach has been virtually abandoned because it tends to result in starting point bias, an effect such that the final WTP amount at the end of the bidding game is systematically related to the initial bid value. Another disadvantage of the approach is that repeated questioning may annoy or tire respondents, causing them to say "yes" or "no" to a stated amount in hopes of terminating the interview.

This paper studies the ability- and willingness- to-pay of informal sector workers who have stopped paying the social health insurance, the Jaminan Kesehatan Nasional (JKN), premium for six months or more in Indonesia. The study uses a mixed methods approach. The authors exploit the responses of 1,709 individuals in 2016 to obtain the willingness-to-pay. Further, they use qualitative interviews to 53 of the respondents to explore in depth which factors explain ability- and willingness-to-pay.

The subject is interesting, and the paper is well written. I also believe the quantitative exercise has been executed correctly. The paper has the strength of combining different datasets and considering the regional differences in ATP, etc. Also, it provides a rich discussion of policy solutions to address the excess of uninsured individuals in Indonesia.

Nevertheless, I have some important caveats that would be addressed for this paper to be publishable at a journal such as PLOS One.

Major:

1. The title does not reflect well the contents of the paper. The study relies on informal sector workers that used to be insured and have failed to pay their premia for 6 months or more. There are 39 million informal sector workers that have never been enrolled and thus are not represented in the sample. The selected sample has characteristics that made them different than those that never have been enrolled. They had decided to pay in the past for whatever reason, while those that never enrolled did not. Therefore, I would recommend changing the title to echo the real content. Also, the authors should argue better why the selection is valid and why - even though the results are necessarily biased, they are useful.

2. The sample is now a bit dated: Being in 2021, to use a survey of 2016 may be a bit of a stretch. One solution would be to provide some statistics to show that the GDP per capita, inflation, etc. has not changed excessively since 2016 (ignoring 2020 as it is a year of complete disruption due to COVID).

3. The method they use to estimate the WTP for health insurance should be better defended. In general, ‘this approach has been virtually abandoned because it tends to result in starting point bias, an effect such that the final WTP amount at the end of the bidding game is systematically related to the initial bid value.’ (Applications of the Contingent Valuation Method in Developing Countries: A Survey, by Anna Alberini, Joseph Cooper, Food and Agriculture Organization of the United Nations, Food & Agriculture Org., 2000 - Business & Economics). It can be argued that this is not a problem in this context, given that there is a point of reference (the premium), but then, the authors need to defend their stance.

4. What dictate the choice of explanatory variables in Table 7. The authors collected much more information from the individuals. The literature has shown that there are more explanatory variables that may influence health insurance decisions (gender of the head of household, number of children in the household, number of elderly, number of health shocks in the last year, etc.). A discussion on the choice of variables included in a regression is needed.

Minor:

1. The subsection called ATP and WTP could be written better to improve its readability. At the moment, the choice of style and formatting make it somewhat cumbersome.

2. The references are outdated. There are a number of articles on WTP for health insurance of informal sector workers in low- and middle- income countries published since 2017.

6. PLOS authors have the option to publish the peer review history of their article (what does this mean?). If published, this will include your full peer review and any attached files.

Reviewer #1: No

Reviewer #2: No

---

## [Author Response · Author response to Decision Letter 0]

25 Mar 2021

The authors would like to thank the reviewers for their thorough reading of the paper, useful comments and constructive suggestions, which have certainly helped in improving the paper.

We checked the PLOS style requirements, and have included a copy of our questionnaires and interview guide in both Indonesian and English as supporting information. Inspired by the COREQ, we have added some more information about those who conducted the qualitative interviews. 

Reviewer #1: This paper addressed one of very critical issues of the health insurance coverage among informal sector from a unique angle. The topic and results of this study could be very valuable for JKN development as well as the development of health insurance schemes in other LMICs.

Here are a few recommendations for the potential improvement.

1. Can author add the sections regarding the literature review and theories related to this study? ATP and WTP have been studied intensively. It would be great that authors can show how these existing knowledges guide the design and analysis of this study. For example, we understand that people’s health state will be an important factor affecting their WTP and actual enrollment of insurance significantly.

Thank you for this comment. We agree with this suggestion and have added several recent articles to the Introduction. While this is not an extensive literature review, the research design has been informed by our understanding of the relevant literature, and the aims of the implementation of JKN in 2014.

2. At the end of the paper, authors made some very useful policy recommendations. It would be great if authors can make better connections between the results of study and these recommendations, with the considerations of feasibilities of their implementation in JKN.

Thank you for this suggestion. In the Discussion, we have explicitly tried to connect each of the recommendations to the results that we present. To keep the formulation of the recommendations clear and concise, we prefer not to further elaborate on these given the space already allocated to recommendations in the Discussion.

3. In line 124, It is stated that “ATP was calculated as five percent of a household’s non- subsistence expenditure, i.e., less the amount spent on food”; while in line 127, it is stated that “it is still half of the most stringent measure of catastrophic health expenditure”. To my knowledge, these two are two different measurements, and please make sure their compatibilities.

Many thanks for pointing this out. We have tried to clarify the text. The reference to catastrophic health expenditure is related to the five percent cut off. The line in 127 is justifying the selection of a five percent cut off: as well as being a ‘typical’ percentage of household income spent on healthcare in LMICs, it is also less than half the most stringent estimation of catastrophic health expenditure (i.e., 10% of income used on healthcare). This section has been edited to clarify this meaning.

Reviewer #2: Referee Report PONE-D-30634

Ability and willingness to pay for National Health Insurance in Indonesia: a study of the informal sector.

The iterative bidding approach (Randall et al., 1974) starts by querying individuals at some initial dollar value and keeps raising (or lowering) the value until the respondent declines (accepts) to pay. This final dollar amount is interpreted as the respondent's WTP. However, this approach has been virtually abandoned because it tends to result in starting point bias, an effect such that the final WTP amount at the end of the bidding game is systematically related to the initial bid value. Another disadvantage of the approach is that repeated questioning may annoy or tire respondents, causing them to say "yes" or "no" to a stated amount in hopes of terminating the interview.

This paper studies the ability- and willingness- to-pay of informal sector workers who have stopped paying the social health insurance, the Jaminan Kesehatan Nasional (JKN), premium for six months or more in Indonesia. The study uses a mixed methods approach. The authors exploit the responses of 1,709 individuals in 2016 to obtain the willingness-to-pay. Further, they use qualitative interviews to 53 of the respondents to explore in depth which factors explain ability- and willingness-to-pay.

The subject is interesting, and the paper is well written. I also believe the quantitative exercise has been executed correctly. The paper has the strength of combining different datasets and considering the regional differences in ATP, etc. Also, it provides a rich discussion of policy solutions to address the excess of uninsured individuals in Indonesia.

Nevertheless, I have some important caveats that would be addressed for this paper to be publishable at a journal such as PLOS One.

Major:

1. The title does not reflect well the contents of the paper. The study relies on informal sector workers that used to be insured and have failed to pay their premia for 6 months or more. There are 39 million informal sector workers that have never been enrolled and thus are not represented in the sample. The selected sample has characteristics that made them different than those that never have been enrolled. They had decided to pay in the past for whatever reason, while those that never enrolled did not. Therefore, I would recommend changing the title to echo the real content. Also, the authors should argue better why the selection is valid and why - even though the results are necessarily biased, they are useful.

We agree with the reviewer and have adapted the title. The title has been changed to represent the main research objective and specify the study focuses on enrollees and not informal sector workers in general.

2. The sample is now a bit dated: Being in 2021, to use a survey of 2016 may be a bit of a stretch. One solution would be to provide some statistics to show that the GDP per capita, inflation, etc. has not changed excessively since 2016 (ignoring 2020 as it is a year of complete disruption due to COVID).

We have added a paragraph in the Discussion on the study’s limitations and mentioned the small changes in real GDP per capita since the survey’s implementation give us little reason to believe the interpretation of the results would change substantially.

3. The method they use to estimate the WTP for health insurance should be better defended. In general, ‘this approach has been virtually abandoned because it tends to result in starting point bias, an effect such that the final WTP amount at the end of the bidding game is systematically related to the initial bid value.’ (Applications of the Contingent Valuation Method in Developing Countries: A Survey, by Anna Alberini, Joseph Cooper, Food and Agriculture Organization of the United Nations, Food & Agriculture Org., 2000 - Business & Economics). It can be argued that this is not a problem in this context, given that there is a point of reference (the premium), but then, the authors need to defend their stance.

Thank you for pointing this out, and for your constructive suggestions. In the above-mentioned paragraph on the study’s limitations, we have also explained the grounding of the bidding game in a real-world value (i.e., the premium) which makes the interpretation of accepted bid value less spurious.

4. What dictate the choice of explanatory variables in Table 7. The authors collected much more information from the individuals. The literature has shown that there are more explanatory variables that may influence health insurance decisions (gender of the head of household, number of children in the household, number of elderly, number of health shocks in the last year, etc.). A discussion on the choice of variables included in a regression is needed.

We have tried to make this clearer and provide some information on the choice of variables. A sentence has been added to the Data Entry and Analysis sub-section which explains how the regressions were built, based on hypothesized associations between personal characteristics and WTP, whereupon the model was refined by removal of non-significant predictors.

Minor:

1. The subsection called ATP and WTP could be written better to improve its readability. At the moment, the choice of style and formatting make it somewhat cumbersome.

Many thanks for this comment. A few sentences in these subsection have been re-written to improve clarity.

2. The references are outdated. There are a number of articles on WTP for health insurance of informal sector workers in low- and middle- income countries published since 2017.

As mentioned in our response to Reviewer One’s comment, we have added several recent articles to the Introduction.

---

## [Decision Letter · Decision Letter 1]

21 May 2021

Why did informal sector workers stop paying for health insurance in Indonesia? Exploring enrollees’ ability and willingness to pay

PONE-D-20-30634R1

Dear Dr. Kok,

We’re pleased to inform you that your manuscript has been judged scientifically suitable for publication and will be formally accepted for publication once it meets all outstanding technical requirements.

Kind regards,

David Hotchkiss

Academic Editor

PLOS ONE

Additional Editor Comments (optional):

Reviewers' comments:

Reviewer's Responses to Questions

**Comments to the Author**

1. If the authors have adequately addressed your comments raised in a previous round of review and you feel that this manuscript is now acceptable for publication, you may indicate that here to bypass the “Comments to the Author” section, enter your conflict of interest statement in the “Confidential to Editor” section, and submit your "Accept" recommendation.

Reviewer #1: All comments have been addressed

Reviewer #2: All comments have been addressed

2. Is the manuscript technically sound, and do the data support the conclusions?

Reviewer #1: Yes

Reviewer #2: Yes

3. Has the statistical analysis been performed appropriately and rigorously? 

Reviewer #1: Yes

Reviewer #2: Yes

4. Have the authors made all data underlying the findings in their manuscript fully available?

Reviewer #1: Yes

Reviewer #2: Yes

5. Is the manuscript presented in an intelligible fashion and written in standard English?

Reviewer #1: Yes

Reviewer #2: Yes

6. Review Comments to the Author

Reviewer #1: (No Response)

Reviewer #2: I think the comments have been addressed satisfactorily. The authors have reflected the suggestions by the reviewers and changed the paper accordingly, including the title.

7. PLOS authors have the option to publish the peer review history of their article (what does this mean?). If published, this will include your full peer review and any attached files.

Reviewer #1: No

Reviewer #2: No

---

## [Editor Report · Acceptance letter]

27 May 2021

PONE-D-20-30634R1 

Why did informal sector workers stop paying for health insurance in Indonesia? Exploring enrollees’ ability and willingness to pay 

Dear Dr. Kok:

I'm pleased to inform you that your manuscript has been deemed suitable for publication in PLOS ONE. Congratulations! Your manuscript is now with our production department. 

Kind regards, 

on behalf of

Dr. David Hotchkiss 

Academic Editor

PLOS ONE